# A 256 × 256 LiDAR Imaging System Based on a 200 mW SPAD-Based SoC with Microlens Array and Lightweight RGB-Guided Depth Completion Neural Network

**DOI:** 10.3390/s23156927

**Published:** 2023-08-03

**Authors:** Jier Wang, Jie Li, Yifan Wu, Hengwei Yu, Lebei Cui, Miao Sun, Patrick Yin Chiang

**Affiliations:** 1State Key Laboratory of ASIC and System, Fudan University, Shanghai 201203, China; 21112020109@m.fudan.edu.cn (J.W.); 18112020006@fudan.edu.cn (M.S.); 2College of Electronics and Information Engineering, Tongji University, Shanghai 201804, China

**Keywords:** LiDAR, 3D imaging, system-on-chip, microlens array, neural network, RGB-guided, depth completion

## Abstract

Light detection and ranging (LiDAR) technology, a cutting-edge advancement in mobile applications, presents a myriad of compelling use cases, including enhancing low-light photography, capturing and sharing 3D images of fascinating objects, and elevating the overall augmented reality (AR) experience. However, its widespread adoption has been hindered by the prohibitive costs and substantial power consumption associated with its implementation in mobile devices. To surmount these obstacles, this paper proposes a low-power, low-cost, single-photon avalanche detector (SPAD)-based system-on-chip (SoC) which packages the microlens arrays (MLAs) and a lightweight RGB-guided sparse depth imaging completion neural network for 3D LiDAR imaging. The proposed SoC integrates an 8 × 8 SPAD macropixel array with time-to-digital converters (TDCs) and a charge pump, fabricated using a 180 nm bipolar-CMOS-DMOS (BCD) process. Initially, the primary function of this SoC was limited to serving as a ranging sensor. A random MLA-based homogenizing diffuser efficiently transforms Gaussian beams into flat-topped beams with a 45° field of view (FOV), enabling flash projection at the transmitter. To further enhance resolution and broaden application possibilities, a lightweight neural network employing RGB-guided sparse depth complementation is proposed, enabling a substantial expansion of image resolution from 8 × 8 to quarter video graphics array level (QVGA; 256 × 256). Experimental results demonstrate the effectiveness and stability of the hardware encompassing the SoC and optical system, as well as the lightweight features and accuracy of the algorithmic neural network. The state-of-the-art SoC-neural network solution offers a promising and inspiring foundation for developing consumer-level 3D imaging applications on mobile devices.

## 1. Introduction

The SPAD-based solid-state LiDAR systems, exhibiting a broad array of applications, outperform traditional rotating LiDAR and microelectromechanical systems (MEMS) LiDAR. This superiority arises from their remarkable mechanical stability, high-performance characteristics, and cost-effectiveness. In contrast to indirect time-of-flight (iToF) systems [1,2], which suffer from limited measurement distances (<20 m), substantial power consumption (tenfold that of direct time-of-flight, dToF), and convoluted calibration algorithms, the dToF imaging system offers a more promising solution. By combining the single-photon sensitivity of SPADs with the picosecond-level temporal resolution of TDCs, dToF systems enable long-range object measurement and centimeter-level depth resolution [3]. Consequently, dToF establishes itself as the predominant technological trajectory for the forthcoming generation of 3D imaging. The capability of measuring depth enables solid-state LiDAR to excel in numerous applications, from floor-sweeping robots and facial recognition for consumers to autonomous driving and 3D building modeling in the industrial domain [4]. As LiDAR technology advances towards more compact system designs, the increased SPAD imaging sensor pixels and depth–RGB image fusion are anticipated to become primary issues to addressed.

Three-dimensional depth imaging employing SPAD technology currently faces certain restrictions, including low spatial resolution, suboptimal effective optical efficiency (i.e., fill factor), and elevated costs. These limitations also impede the expansive array of applications for dToF LiDAR systems. Although recent research advancements indicate that SPAD arrays can achieve QVGA (320 × 240) resolutions and even higher (512 × 512), with prospective advancements targeting Video Graphics Array (VGA, 640 × 480) resolutions, the pixel number remains markedly inferior to traditional CMOS Image Sensor (CIS) technology [5,6,7,8]. Apple LiDAR (256 × 192) and Huawei P30 (240 × 180) exemplify the potential of dToF technology in the consumer market, yet the resolution disparity persists. To address this challenge, the 3D stacking technique has been proposed, which positions the back-illuminated SPAD array on the top tier and the readout control circuits on the bottom tier [9]. This configuration allows researchers to enhance the resolution to 512 × 512 [10]. However, the considerable costs associated with this technique have impeded its development and subsequent applications. At present, no effective solutions have been identified to surmount this obstacle in the realm of circuit design.

The inherent limitations of consumer-grade LiDAR technology, specifically its low resolution and high cost, have long been recognized within the industry. As 3D sensing technology advances, consumer-grade RGB-D cameras (e.g., Intel RealSense, Microsoft Kinect) have gained popularity owing to their ability to capture color images and depth information simultaneously, then recovering 3D scenes at a lower cost. Motivated by this development, researchers have explored RGB-guided depth-completion algorithms to reconstruct depth–density maps from sparse depth data obtained by dToF depth sensors and color images captured by RGB cameras, with the goal of predicting low-cost LiDAR-generated high-resolution scenes [11]. The authors [12] introduce a normalized convolution layer that enables unguided scene depth completion on highly sparse data using fewer parameters than related techniques. Their proposed method treats validity masks as a continuous confidence field and presents a new criterion for propagating confidences between CNN layers. This approach allows for the generation of pointwise continuous confidence maps for the network output. The authors [13] propose convolutional spatial propagation networks (CSPNs), which demonstrate greater efficiency and accuracy in depth estimation compared to previous state-of-the-art propagation strategies, without sacrificing theoretical guarantees. Liu et al. [14] proposed an architecture, the differentiable kernel regression network, which consists of a CNN network that learns steering kernels to replace hand-crafted interpolation for performing the coarse depth prediction from the sparse input.

Gaussian beams play a significant role in various applications, including LiDAR, optical communication, and laser welding [15]. However, in flash LiDAR and laser TV projection, it is essential to homogenize Gaussian beams into flat-topped beams. Common techniques to generate flat-top beams include MLA, diffractive optics (DOE), and free-form mirror sets. Among these technologies, MLA-based beam homogenizers have garnered considerable interest, particularly in compact consumer devices, owing to their unique properties. An MLA can divide a non-uniform laser into multiple beamlets, which can subsequently be superimposed onto a microdisplay with the assistance of an additional lens [16]. As a result, MLA diffusers display independence from the incident intensity profile and a wide spectral range. In 2016, Jin et al. [17] proposed enhancing homogeneity in the homogenizing system by substituting the first MLA with a free-form MLA. Each free-form surface within the MLA introduced suitable aberrations in the wavefront, redistributing the beam’s irradiance. In the same year, Cao et al. [18] presented a laser beam homogenization method that employed a central off-axis random MLA. By adjusting the off-axis quantity of the center, the MLA’s periodicity was disrupted, eliminating the periodic lattice effect on the target surface. In 2019, Liu et al. [19] designed a continuous profile MLA featuring sublenses with random apertures and arrangements. This innovation facilitated a breakthrough in beamlet coherence, achieving a simulated uniformity of 94.33%. Subsequently, in 2020, Xue et al. [20] proposed a monolithic random MLA for laser beam homogenization. During this homogenization process, the coherence between sub-beams was entirely disrupted, resulting in a homogenized spot with a high energy utilization rate.

In an effort to enhance the resolution of low-pixel depth sensors and broaden the application scope of sparse-pixel, low-cost ranging sensors, we propose a SPAD-based LiDAR with a micro-optics device and RGB-guided depth-complemented neural network to upsample the resolution from 8 × 8 to 256 × 256 pixels as shown in Figure 1. This lightweight, SPAD-based dToF system is ideal for LiDAR applications. In this work, the developed sparse-pixel SoC was fabricated with the 130 nm BCD process. The high photon detection probability (PDP) of the SPAD and picosecond TDC ensure that the system effectively achieves millimeter-level measurement accuracy in indoor scenarios. The sensor integrates a 16 × 16 SPAD array, which can be divided into 2 × 2 areas at high frame rates for ranging or 8 × 8 at low frame rates for imaging. Subsequently, we engineered an optical system to facilitate imaging at the hardware level. Micron-scale random MLAs were employed on the transmitter to homogenize the vertical-cavity surface-emitting laser (VCSEL) array’s Gaussian beam into a flat-topped source with a 45° FOV. Free-form lenses were applied at the receiver to align the SPAD array with a 45° FOV and enhance the resolution for sparse imaging using the 8 × 8 array. To meet consumer-grade imaging requirements, an RGB-guided depth complementation neural network was implemented on an NVIDIA Jetson Xavier NX (Deep Learning Accelerator) and PC, improving the resolution from 8 × 8 to 256 × 256 pixels (QVGA level). This cost-effective, lightweight imaging system has a wide range of applications in distance measurement, basic object recognition, and simple pose recognition. To validate the system, we conducted ranging verification on the SPAD-based SoC LiDAR, as well as FOV and resolution tests on micro-optical modules. Our self-developed RGB-guided depth completion neural network upsampled and complemented 8 × 8 depth information by ×32 to map the real world. The paper is structured as follows: Section 2 presents the SoC implementation and key functional component design; Section 3 discusses the optical system simulation and MLA design; Section 4 introduces the proposed RGB-to-depth completion architecture; Section 5 reports the results and discussion of Section 2, Section 3 and Section 4; and Section 6 provides conclusions and future perspectives.

## 2. SOC Implementation

Traditional LiDAR systems rely on a ranging framework in which the transmitter and receiver operate as separate entities. A short laser pulse is illuminated by the transmitter, which then undergoes diffuse reflection upon encountering an object before returning to the receiver. Here, the SPAD captures the corresponding photons and subsequently produces photon counts. This signal is conveyed through a multiplexed TDC, ultimately reaching the digital signal processor (DSP) for processing and generation of a time-dependent histogram. However, the conventional architecture is encumbered by several limitations, such as the challenging task of achieving picosecond-level pulsing and the system’s inherently poor signal-to-noise ratio (SNR) [21,22]. We propose a highly integrated SoC based on a 130 nm BCD process, which integrates a SPAD readout circuit, TDC, charge pump circuit for high-voltage generation, and ranging algorithm processor, as illustrated in Figure 2. This design offers superior temporal resolution and SNR compared to conventional architectures, as well as significant cost advantages. The ranging SoC integrates two SPAD pixel arrays; namely, the signal SPAD array is utilized for depth imaging, while the reference SPAD array is employed for offset calibration between the sensor and cover glass. Figure 2 illustrates the signal path for both pixel arrays within the ranging SoC. The high voltage of all SPADs is generated in charge pump, which is linked to the high-voltage drain drain (HVDD). Each region of the SPAD array corresponds to a TDC output, and the output from the TDC is subsequently stored in static random-access memory (SRAM) for histogram statistics. The SoC is interconnected with an external laser driver using low-voltage differential signaling (LVDS), enabling the adjustment of laser pulse width and power. Additionally, the SoC integrates a microcontroller unit (MCU) capable of performing on-chip operations, including the implementation of the ranging algorithm processor.

Figure 3a illustrates the SPAD pixel circuit. The cathode of the SPAD is connected to a high voltage bias, which is provided via the HVDD connection. The passive quench circuit implemented by a thick-gate transistor device enables the overbias voltage of the SPAD to reach up to 3.3 V. Following level conversion, the output signal of the SPAD pixel is reduced to a low voltage range compatible with the thin gate transistor. The detection threshold of the SPAD avalanche signal is determined by the buffer of the SPAD posterior stage. By adjusting the supply voltage of the pixel circuit, the detection threshold of the SPAD avalanche signal can be modified. In Figure 3b, the design of the SPAD readout circuit is presented. Four SPADs are combined into a macropixel using an OR-tree structure. This macropixel configuration addresses the issue of low photon detection efficiency in individual SPADs. The probability of triggering an avalanche signal in this 2 × 2 pixel structure is four times that of a single pixel, and the avalanche signal from each pixel can be output through the OR-tree. Moreover, each SPAD can be individually enabled, allowing for flexible configuration of the SPAD array size.

The design structure of the TDC is shown in Figure 4. It operates by synchronizing the pulse optical signal generated by the laser as the start signal of the TDC, while the SPAD triggers the generation of an avalanche signal, serving as the stop signal. The time interval between these two signals is recorded to output information from the TDC. Initially, a clock counter is employed to measure the number of clocks between the start and end signals, generating a rough count. The resolution of the rough count is determined by the clock frequency, making it challenging to achieve high-frequency rough counts. To address this limitation, a fine-count structure is introduced to improve the frequency of the TDC. The design of the fine count incorporates a multi-phase clock scheme. In this scheme, an on-chip oscillator generates a 10.19 MHz clock source, which serves as the input to a phase-locked loop (PLL). The PLL produces a 234 MHz clock output after 23 frequency multiplications. The clock is directly used for rough counting to obtain the output of TDC<7:3>, and the 234 MHz clock is shifted. Additionally, eight clocks, denoted as P0 to P7, with the same frequency but different phases, are derived from the PLL clock. The phase of P0 aligns with the phase of the PLL output clock. These eight clocks, with the same frequency and different phases, are then utilized for sampling and coding, resulting in the output of TDC<2:0>. Finally, the output of the TDC is fed into the digital circuit for histogram processing. The temporal resolution of the TDC is specified as 300 ps.

## 3. Optical System and MLA

In active imaging applications, particularly for LiDAR systems, it is imperative to optimize not only the receiver components, such as lenses and sensors, but also the transmitter elements and the target object. This optimization process requires an in-depth understanding of various factors, including the laser emission mode (flash or dot matrix) and the physical mechanism governing the object’s reflection. The optical model is constructed by taking into account calculations derived from the optical power at the transmitter, the single-pixel optical power budget at the receiver, and the lens system. This model can be readily extended to encompass the entire array once the illumination pattern is known. In the standard scenario, the illumination pattern along the detector’s field of view in both horizontal and vertical directions is designed to match the FOV angle at the transmitter. Employing complex physics can facilitate the incorporation of the optical model by calculating the optical power density on the target surface.

Figure 5 presents a generic flash LiDAR model, in which the scene of interest is uniformly illuminated by an active light source, typically a wide-angle pulsed laser beam. The coverage area in the target plane is contingent upon the FOV of the beam. All signal counting calculations are predicated on the lens focusing the reflected light energy back to the SPAD array. To maximize the efficiency of returned energy throughout the field of view, it is crucial to ensure a well-matched FOV between the transmitter and sensor. Regarding the returned light, the target is assumed to be a diffuse surface, i.e., a Lambertian reflector, which exhibits consistent apparent brightness to the observer, irrespective of the observer’s viewpoint, and the surface brightness is isotropic. The active imaging system setup is illustrated in Figure 5, with the distance to the target as D, the lens aperture as d, the focal length as f, the sensor height as a, and the area as $A_{sensor}$. The reflected light power, $P_{target}$, emanating from the target is determined by the source power, $P_{source}$(D), and the reflectance, $\rho_{target}$, of the object. The lens component encompasses the f-number (f# = f/d) and FOV. Additionally, the light transmission rate, $\eta_{lens}$, of the lens and filter must be considered. When treating the illumination light as square, the optical power model can be articulated by Equation (Equation 1): (1)$$P_{received}=P_{source}\cdot \rho_{target}\cdot {\left(\frac{d}{2\cdot f}\right)}^{2}\cdot \eta_{lens}\cdot \frac{2\cdot A_{sensor}}{\pi }\cdot {\left(\frac{1}{2\cdot D\cdot tan(FOV/2)}\right)}^{2}$$

Transitioning from the optical power model to a photon counting model proves to be more advantageous for system design and sensor evaluation. Consequently, the photon counting model can be formulated as presented in Equation (Equation 2): (2)$$N_{pulse}=P_{received}\cdot \frac{\lambda }{F_{laser}\cdot h\cdot v}$$
where $F_{laser}$ is the laser repetition frequency as shown in Figure 5, λ is the light source wavelength used, h is Planck’s constant (6.62607004 × $10^{34}$ J· s), and *v* is the speed of light (2.998 × $10^{8}$ m/s). This value defines the total number of photons per laser pulse that reaches the pixel array.

The diffusion beam function of the laser diffuser is primarily achieved through the micro-concave spherical structure etched on its surface. This micro-concave spherical surface acts as a micro-concave lens, and the entire laser diffuser can be considered an array of micro-concave lenses. Essentially, the diffused laser surface light source is a superposition of the surface light sources emitted by each micro-concave lens. It is necessary to homogenize the Gaussian beam into a flat-topped beam to achieve a more uniform projected light source for active imaging as illustrated in Figure 6e. As illustrated in Figure 6d, the laser diffuser diffusion diagram demonstrates the laser incident vertically on the diffuser, with the laser beam diverging due to the influence of the micro-concave lens. This forms a surface light source with a specific diffusion angle, which is associated with the parameters of the micro-concave lens on the diffuser. In accordance with this principle, this paper proposes the preparation of two types of MLAs: the random MLA, illustrated in Figure 6b, suitable for generating a circular homogenized light spot; and the rectangular MLA, presented in Figure 6c, designed to produce a rectangular homogenized light spot.

To elucidate the specific relationship between the characteristic parameters of the micro-concave spherical structure and the diffusion angle, a deductive argument will be presented from the perspective of geometrical optics. Ideally, the overall diffusion angle of the laser beam after traversing the diffuser is equivalent to the diffusion angle of α single sublens. To simplify the analysis of the diffusion angle regarding the diffuser structure parameters, the sublens diffusion process is examined individually. Figure 6a illustrates the diffusion diagram of the sublens, which involves two parameters: the hole P and the radius of curvature R. By employing the lens focal length formula and geometric principles, the set of Equations (3) is derived. By further simplifying the system of equations, Equation (Equation 4) can be obtained.
(3)$$tan\theta =\frac{P}{2f},\left|f\right|=\frac{R}{n-1}$$
(4)$$\theta =arctan\frac{P(n-1)}{2R}$$
where n is the refractive index of the micro-concave lens material and f is the focal length of the micro-concave lens. If the parameters of the other sublenses are known, the diffusion angle of the sublenses can be derived, which in turn yields the diffusion angle of the laser diffuser. Equation (Equation 4) further reveals that the diffusion angle theta is directly proportional to the aperture P of the micro-concave lens and inversely proportional to the radius of curvature R.

## 4. Proposed RGB-Guided Depth Completion Neural Network

The primary aim of our research is to accurately estimate a dense depth map, designated as Ŷ, derived from sparse LiDAR data (Y’) in conjunction with the corresponding RGB image (I) serving as guidance. W denotes the network parameters. The task under consideration can be succinctly expressed through the following mathematical formulation: (5)$$\hat{Y}=F\left(Y’,I;W\right),$$

We implemented an end-to-end network, denoted as F, to effectively execute the depth-completion task in our study. The optimization function can be concisely represented in the subsequent description: (6)$$\hat{W}=\underset{W}{\arg\min}L\left(\hat{Y},Y’;W\right),$$

*L* represents the loss function employed for the purpose.

In our proposed methodology, the network is divided into two distinct components: (1) a depth-completion network designed to generate a coarse depth map from the sparse depth input, and (2) a refinement network aimed at optimizing sharp edges while enhancing depth accuracy. We introduce an encoder–decoder-based CNN network for estimating the coarse depth map, as previously demonstrated in [13,14]. However, the aforementioned networks encounter two primary challenges. First, networks such as [12,14] employ complex architectures to boost accuracy; nevertheless, their proposed networks are incompatible with most CNN accelerators or embedded DLAs. Second, networks like [13] utilize ResNet-50 [23] or other similar networks as the CNN backbone, resulting in a substantial number of parameters (~240 M) and posing implementation challenges on embedded hardware.

To make the network easier to implement on the embedded systems, we explored two optimization strategies:We incorporated conv-bn-relu and conv-bn architectures, which are supported by the majority of CNN accelerators and embedded NPUs.To reduce the number of parameters, we adopted depthwise separable convolution, as presented in [24], as the foundational convolution architecture.

### 4.1. Depthwise Separable Convolution

The architecture of depthwise separable convolution is illustrated in Figure 7. In contrast to the conventional 3 × 3 2D convolution, depthwise separable convolution employs a 3 × 3 group convolution, referred to as depthwise convolution, and a 1 × 1 standard convolution, known as pointwise convolution.

For a standard convolution with a filter K of dimensions $D_{k}\times D_{k}\times M\times N$ applied to an input feature map F of size $D_{f}\times D_{f}\times M$, the total number of parameters can be computed as:(7)$$D_{k}\times D_{k}\times D_{f}\times D_{f}\times M\times N,$$

In the case of depthwise separable convolution, the total parameters are calculated as follows:(8)$$D_{f}\times D_{f}\times D_{k}\times D_{k}\times M+M\times N\times D_{k}\times D_{k},$$

Considering that the kernel size $D_{k}$ for our network is 3 and the output channel N is considerably larger than $D_{k}\times D_{k}$, the standard convolution possesses a significantly higher number of parameters in comparison to depthwise separable convolution.

### 4.2. Network

Different from the standard CNN network, we substitute the standard 3 × 3 convolution with our depthwise separable convolution. Our network architecture is depicted in Figure 8. Within the depth-completion network, we employ an U-net [25] network as our backbone. The encoder consists of five stages, each containing two types of depthwise separable convolution blocks. One block performs downsampling by a factor of 2× using a stride of 2, while the other extracts features with a stride of 1. The decoder also consists of five stages. In the first four stages, feature maps are upsampled by 2×, followed by a convolution operation with a stride of 1. These maps are then concatenated with the feature maps from the corresponding encoder stage outputs and processed by two subsequent convolution blocks with a stride of 1. The final stage comprises an upsampling layer and a single convolution block with a stride of 1. For the refinement network, we adopted the architecture from [12], which has been demonstrated to possess the advantages of being lightweight and exhibiting high performance.

### 4.3. Loss Function

We trained our network in two steps: In Step 1, we trained our network using the L1 error as our loss function. The loss function can be described as follows:(9)$$L_{1}=\frac{1}{M}\sum_{i,j}\left(Y_{i,j}>0\right)\left|{\hat{Y}}_{i,j}-Y_{i,j}\right|$$
where $Y_{i,j}$ denotes the ground truth at pixel (i,j), ${\hat{Y}}_{i,j}$ represents the predicted depth at pixel (i,j), and *M* stands for the total number of pixels with values greater than 0.

Upon completing the training of the network, we proceeded to train the entire network using a new loss function to optimize the boundary region performance, which can be described as follows:(10)$$L=w_{1}L_{1}+w_{2}L_{2}+w_{3}L_{gard}$$

In this equation, $w_{1}$, $w_{2}$, and $w_{3}$ are hyperparameters. $L_{1}$ is defined similarly to Equation (Equation 5), $L_{2}$ represents the masked mean squared error (MSE), and $L_{gard}$ stands for the gradient error. To achieve improved boundary performance, we incorporated depth gradient information into our loss function, following [26]. The gradient loss function, $L_{grad}$, penalizes the disagreement of depth edges in both vertical and horizontal directions.

## 5. Results and Discussion

This chapter presents a systematic analysis of the verification results obtained for the hardware, optics, and depth imaging networks. The initial focus is on the verification of the SoC and optical modules. Furthermore, a comprehensive systematic verification of the RGB-guided depth reconstruction imaging system was conducted, resulting in an improved efficiency of the hardware and algorithm combination. Ultimately, this approach leads to the development of a lightweight and cost-effective system for depth imaging of indoor scenes.

### 5.1. System of Hardware

To assess the low power characteristics of the LiDAR SoC, tests were conducted on the total power consumption of the SPAD-based SoC and the externally driven laser-emitting component. The SPAD-based SoC, operating at a power supply voltage of 3.3 V and an RMS operating current of 12 mA, demonstrated a total power consumption of approximately 43 mW. When in the luminous state, the externally driven laser-emitting component operated at a supply voltage of 6 V and an RMS current of 25 mA, resulting in a total power consumption of 150 mW. Consequently, the overall power consumption of the LiDAR system was approximately 200 mW.

To accommodate the diverse scenarios and complex applications in indoor environments, the SoC based on micro-structured optical element packaging was tested using a Lambertian reflector plate with reflectivities of 85% and 10%, respectively. As illustrated in Figure 9a,b, with a 85% Lambertian reflector plate and an indoor lighting environment of 10 klux, the SOC achieves a maximum measurement distance of 6 m and a minimum measurement distance of 0.2 m while maintaining at least 10 mm precision within the measurement range. Under the same conditions, the SoC with a 10% Lambertian reflector plate exhibits a maximum measurement distance of 4 m and a minimum measurement distance of 0.2 m, demonstrating at least 15 mm precision throughout the measurement range. These two scenarios encompass mainstream indoor applications and fully showcase the exceptional performance of SoCs packaged with micro-structured optical components at the hardware level. Figure 9c illustrates the curve that compares the simulated photon count with the tested photon counts versus the actual detection distance, providing a demonstration of the crucial impact of the preceding optical system modeling. As illustrated in Figure 9e,f, the micro-structured optical elements encapsulated in the SoC can perform 8 × 8 sparse depth imaging of objects (10% reflectance) at a distance of approximately 3 m, without any loss of depth information at the edges. This feature proves to be highly beneficial for subsequent depth-completion algorithms.

Monte Carlo analysis and simulation were conducted to evaluate the performance of randomly distributed microlens array (MLA) homogenizers. In the optical simulation model, the diffusion sheet dimensions are 4 × 4 mm with a 0.5 mm thickness. To ensure consistency between the diffusion half-angle a of the effective illumination area calculated by fluctuating optics and the micro-structural characteristics, the simulated microlens aperture P was set to 40.4 µm, the radius of curvature R to 50 µm, and the selected material was polymethyl methacrylate (PMMA) with a dielectric refractive index n = 1.5. The detector was positioned 100 mm downstream of the diffusion sheet, material absorption characteristics were neglected, and the diffusion effect is depicted in Figure 9c. From Figure 9c, it can be observed that the energy distribution of the surface light source, obtained after the laser beam’s diffusion through the diffuser, remains strong at the center and weak at the edges. However, the illumination distribution curve is more uniform compared to the undiffused laser. Furthermore, an energy utilization rate of up to 90% can be derived from the total power captured on the detector. A diffusion half-angle of 22° in the full illumination region can also be determined from the coordinates of the zero position of the detector’s Y-axis, which aligns with the geometric optical description.

### 5.2. Comparison of Network Performance

#### 5.2.1. Dataset

To adapt our method for indoor hardware systems, we evaluate its performance using the NYUv2 dataset [27], which consists of RGB and depth images gathered from 464 distinct indoor scenes as shown in Figure 10. We employed the official split, utilizing 47 k images for training and 654 images for evaluation. For the purpose of comparing our network’s performance with existing methods, we adopted data processing techniques akin to those described in [13,28]. The RGB-D images were subjected to downsampling and center-cropping, resulting in a resolution of 304 × 228. Moreover, we randomly sampled 500 depth points to serve as the sparse depth input for our network. This strategy ensures a fair comparison with other methods while preserving the integrity of our evaluation.

#### 5.2.2. Metrics

We adopted the same metrics and used their implementation in [28]. Given ground truth depth Y and predicted depth $\hat{Y}$, the metrics include:RMSE:
(11)$$\sqrt{\frac{1}{M}\sum_{i,j}\left(Y_{i,j}>0\right){\left({\hat{Y}}_{i,j}-Y_{i,j}\right)}^{2}}$$Abs Rel:
(12)$$\frac{1}{M}\sum_{i,j}\left(Y_{i,j}>0\right)\left|{\hat{Y}}_{i,j}-Y_{i,j}\right|/Y_{i,j}$$$\delta_{t}$: % of $Y_{i,j}$, s.t.
(13)$$max\left(\frac{{\hat{Y}}_{i,j}}{Y_{i,j}},\frac{Y_{i,j}}{{\hat{Y}}_{i,j}}\right)<t,t\in \left(1.10,1.25,1.25^{2},1.25^{3}\right)$$

#### 5.2.3. Comparison

Table 1 presents a comparison of the qualitative results of our network and four state-of-the-art methods [4,12,13,27]. As illustrated, our network achieves superior performance while preserving a reduced number of parameters (~1 M). This enhanced efficiency is attributed to the implementation of depthwise separable convolutions and the proposed CNN-based lightweight depth completion network. Consequently, our network can be readily deployed on CNN accelerators and neural processing units (NPUs). A noteworthy feature of our network is its enhanced capacity to distinguish boundary regions, which is primarily owing to the employment of a gradient loss function.

### 5.3. Network Implementation on Hardware System

As demonstrated in Section 5.2, our network exhibits state-of-the-art performance in RGB-guided depth completion tasks as shown in Table 2. For instance, the network achieves a root mean square error (RMSE) of 0.116 m on the NYUv2 training set, allowing for more detailed identification at the boundaries. In order to implement the proposed network within our hardware system, the 8 × 8 depth sensor was aligned with the RGB data (256 × 256). On the network side, we adapted the network input according to the hardware’s RGB size and the aligned depth map coordinates, retrained the neural network, and integrated it with the hardware. The resulting completed depth map is illustrated in Figure 11, where the boundaries and shapes are clearly discernible, meeting the fundamental requirements for consumer-level imaging applications.

## 6. Conclusions

In this work, we propose a LiDAR system based on SPAD integrating micro-optical devices, incorporating RGB-guided 8 × 8 depth completion to 256 × 256 pixels with a lightweight neural network. To verify the effectiveness, we present a ranging SoC based on a 130 nm BCD process. The integrated SPAD and TDC enable the system to achieve millimeter-level measurement accuracy in indoor environments. The 16 × 16 pixel SPAD array can be divided into 2 × 2 regions for ranging at high frame rates, or 8 × 8 for sparse imaging at lower frame rates with an energy efficiency of 200 mW. Next, a micro-optical system is proposed to make imaging feasible at the hardware level. A micrometer-scale random MLA is used at the transmitting end to homogenize and expand the Gaussian beams from the VCSEL array into a flat-top light source with a 45° FOV. A freeform lens is utilized in the receiver, allowing the alignment of the SPAD array with the 45° FOV illumination and achieving 8 × 8 array sparse imaging at the optical level. To further increase the resolution to meet the requirements of consumer-level imaging, an RGB-guided depth complementation neural network was implemented on an NVIDIA Jetson Xavier NX (Deep Learning Accelerator) and a PC, reaching a 256 × 256 resolution that matches QVGA standards. The low-cost, lightweight depth imaging system has widespread applications in distance measurement, simple object recognition, and basic pose recognition technology fields.

## Figures and Tables

**Figure 1 sensors-23-06927-f001:**
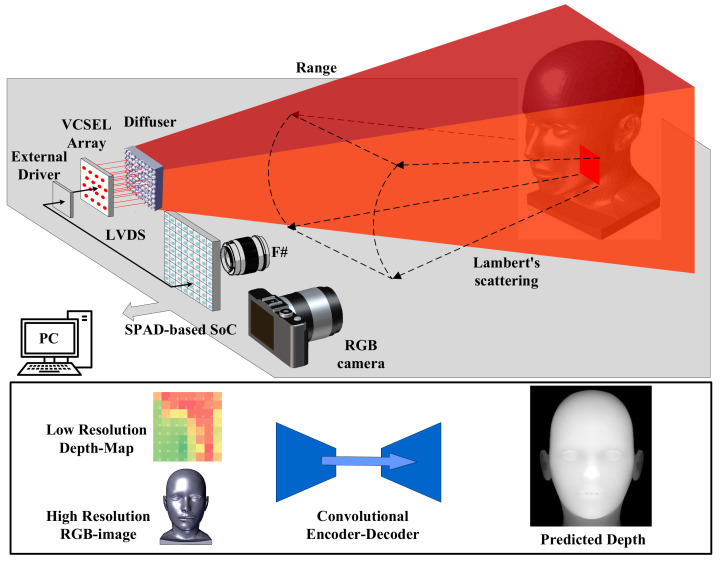
The proposed low-cost, lightweight SPAD-based SoC with micro-lens array and RGB-guided 256 × 256 depth completion for 3D LiDAR imaging.

**Figure 2 sensors-23-06927-f002:**
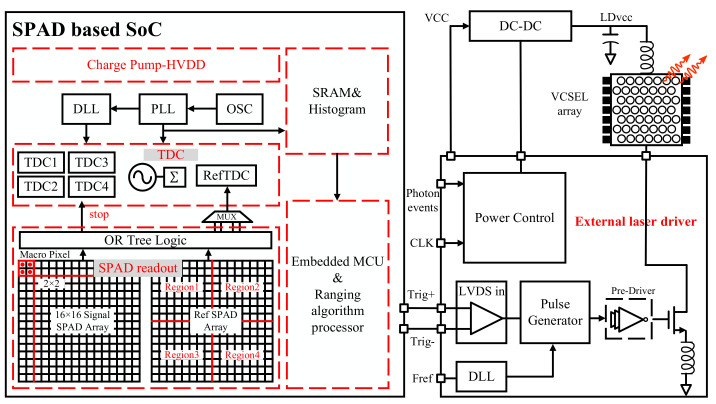
Architecture of dToF LiDAR SoC.

**Figure 3 sensors-23-06927-f003:**
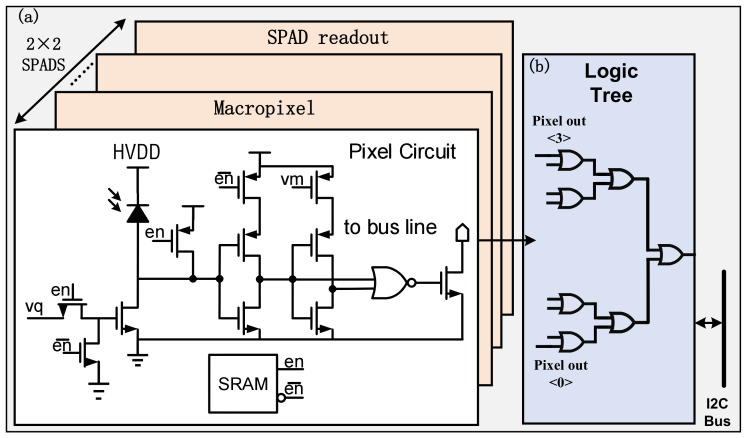
Schematic of SPAD’s readout scheme. (**a**) Pixel circuit. (**b**) Logic of SPAD read out.

**Figure 4 sensors-23-06927-f004:**
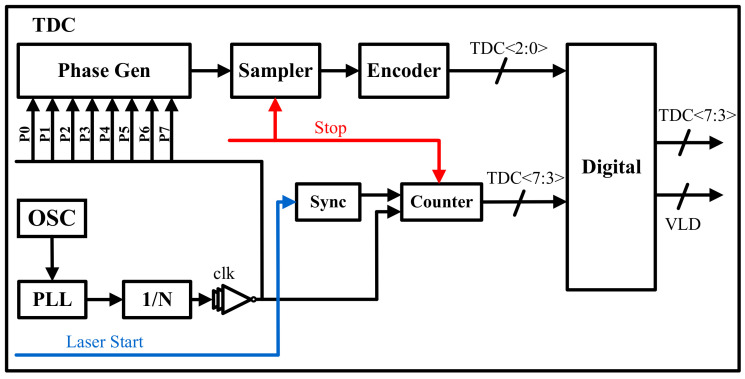
Simplified block diagram of TDC.

**Figure 5 sensors-23-06927-f005:**
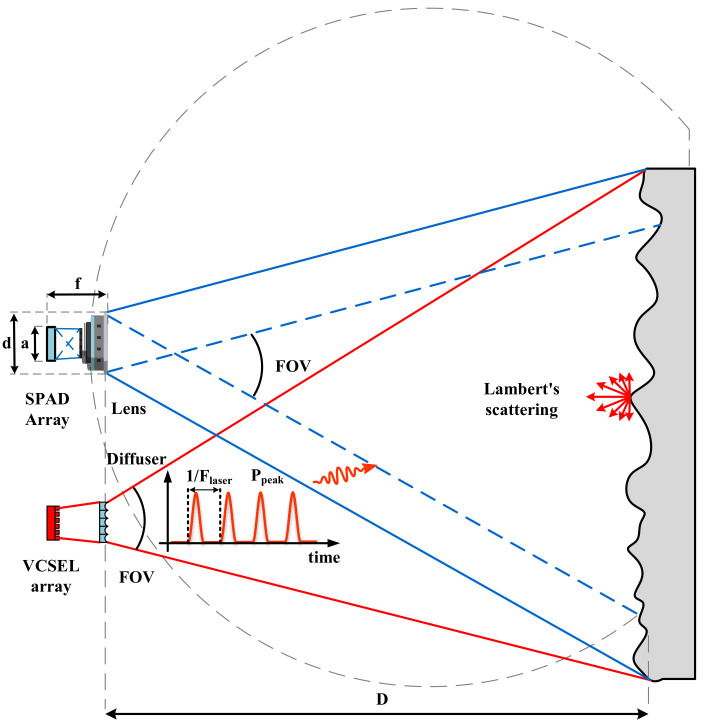
Typical active imaging system model, using a Lambertian reflectance as target.

**Figure 6 sensors-23-06927-f006:**
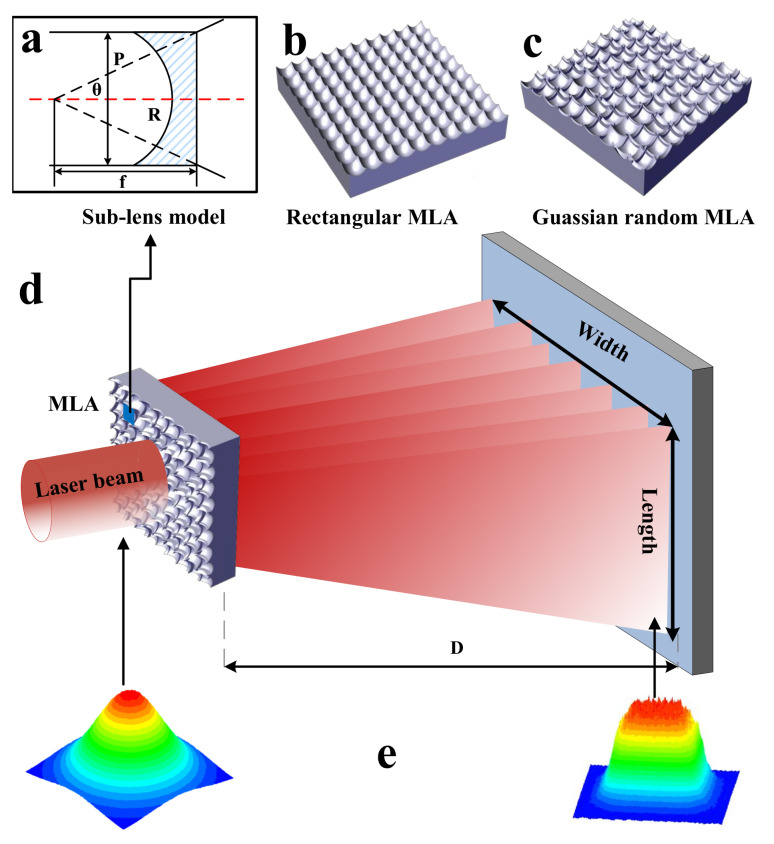
(**a**) Structural parameters of microlens unit; (**b**) rectangular MLA; (**c**) random MLA; (**d**) schematic diagram of random microlens array; (**e**) Gaussian beams to flat-topped beams.

**Figure 7 sensors-23-06927-f007:**
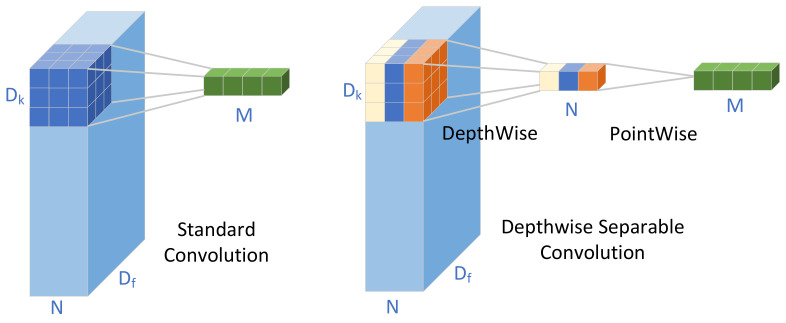
Architecture of depthwise separable convolution.

**Figure 8 sensors-23-06927-f008:**
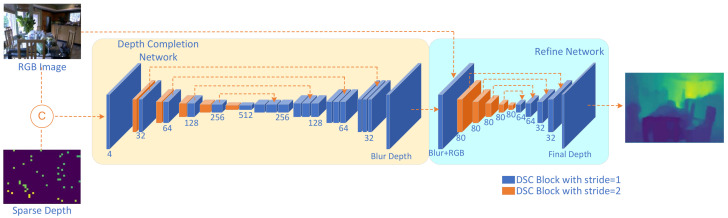
The proposed RGB-guided depth completion network. C is Concat in circle.

**Figure 9 sensors-23-06927-f009:**
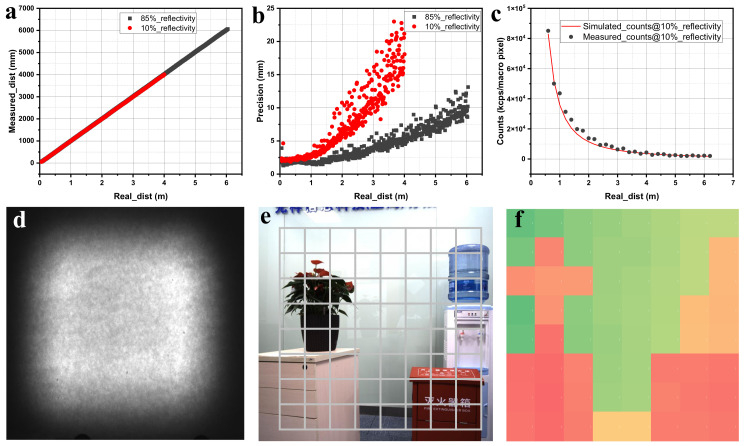
Verification of hardware. (**a**) Distance at 85% and 10% reflectivity test conditions; (**b**) precision at 85% and 10% reflectivity test conditions; (**c**) simulated and tested photon counts versus the actual detection distance; (**d**) the output of intensity distribution on the screen radiated by the MLA diffuser; (**e**) aligned depth sensor’s zones and color image; (**f**) 8 × 8 depth map.

**Figure 10 sensors-23-06927-f010:**
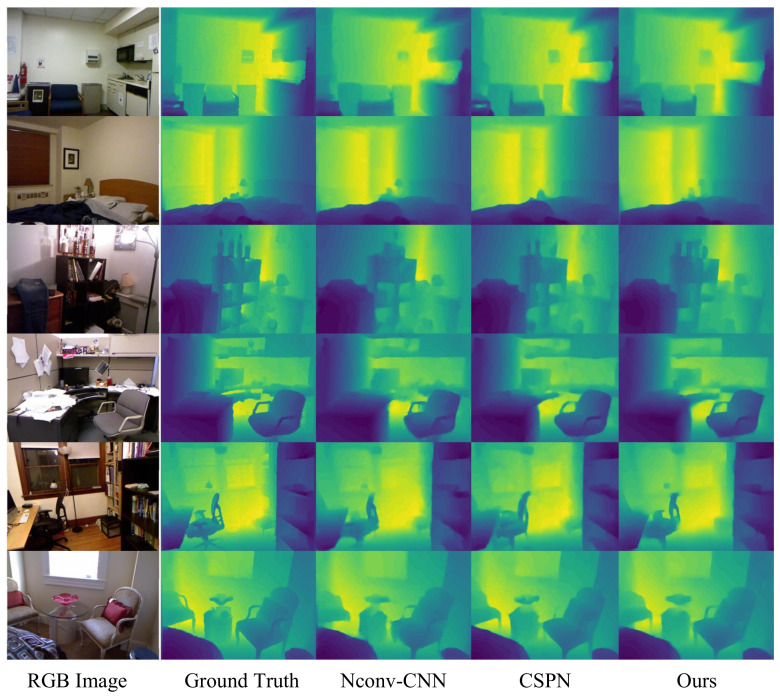
Qualitative Comparison on NYUv2 [27]. From left to right: guidance RGB image, the ground truth depth map, results of Nconv-CNN [12], CSPN [13], and ours.

**Figure 11 sensors-23-06927-f011:**
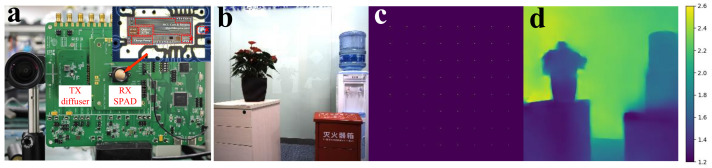
The result of our hardware system. (**a**) RGB–depth completion setup; (**b**) Guidance RGB image; (**c**) the sparse depth map; (**d**) the predicted depth map from our network.

**Table 1 sensors-23-06927-t001:** Specifications of LiDAR System.

Items	This Work	[3]	[29]
SoC and laser power, $P_{avg}$	200 mW	306 mW	2540 mW
Pixel array	8 × 8	240 × 160	252 × 144
Laser wavelength, λ	940 nm	940 nm	940 nm
Repetition frequency, $f_{laser}$	10 MHz	10 MHz	40 MHz
Exposure time	10∼200 ms	-	-
Field-of-view, FOV	45°	63 × 41°	40 × 20°
Imaging resolution	256 × 256	240 × 160	252 × 144

**Table 2 sensors-23-06927-t002:** Summary of essential characteristics of existing RGB-guided methods on the NYU-v2 dataset. For denoting loss functions. We omit the coefficient of each loss term for simplicity.

Method		Error			Accuracy		Parameters
	Rmse	Rel	$\delta_{1.10}$	$\delta_{1.25}$	$\delta_{1.25^{2}}$	$\delta_{1.25^{3}}$	
Sparse-to-Dense [28]	0.230	0.044	92.6	97.1	99.4	99.8	28.4 M
Unet+CSPN [13]	0.117	0.016	97.1	99.2	99.9	100.0	256 M
KernelNet [14]	0.111	0.015	97.4	99.3	99.9	100.0	16.47 M
Nconv-CNN [12]	0.125	0.017	96.7	99.1	99.8	100.0	484 K
Ours	0.116	0.018	96.8	99.3	99.9	100.0	1.07 M

## Data Availability

Data sharing not applicable.

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
