# Peer review of "A 256 × 256 LiDAR Imaging System Based on a 200 mW SPAD-Based SoC with Microlens Array and Lightweight RGB-Guided Depth Completion Neural Network"

_sensors, 2023, doi:10.3390/s23156927_

Round 1
Reviewer 1 Report
- Overall I would like to recommend to accept the publication with minor revisions and language checks
- Some suggestions include the following:
- I suppose that CIS in line 45 means CMOS Image Sensor instead of Contact Image Sensor
- A traditional Lidar architecture is considered to have poor SNR in line 129. Could a reference be added to support that statement and to provide the reader the opportunity to get more information about the traditional approach?
- It is not clear from the descriptions like in line 135 and from the figures 2 and 3 whether there are two separate physical SPAD arrays or whether the same pixel array can be readout in 2 different modes. Some clarification would be useful here to avoid confusion.
- Figure 4 and text below in lines 184 to 169 have some inconsistent parameters such as d2a_tof_clk_div. Probably figure 4 could further be simplified to focus only on relevant signals. Please also provide some numbers when talking about precise time measurements.
- Laser frequency in line 202 needs to be defined, potentially a timing diagram could help.
- Not sure if the relation between equations 10 to 12 are obvious to the general reader.
- Section 5 lines 309 to 323: Please provide dToF system conditions like wavelength, power, exposure time, laser frequency. In this context, how much spurious signal is actually present when talking about an 10klux indoor environment?
- Same paragraph, please skip valuations like "good" precision since this is subjective. Also, comparing the precision with figure 9a, is it actually 15% or 1.5%?
- Ideally the reader would like to see light budget validation in figure 9a. Could a theoretical curve be added to the graph?
- Nice comparison of the network performance KPIs with other algorithms
Reviewer 2 Report
Please refer to the attached pdf.

The Quality of English Language is fine, but the logic of sentences are strange.
I cannot well identify what is the conventional and what is the proposal.
Some typos are found. No space after the period. Some sentences, especially equations, ends with a comma.
Reviewer 3 Report
This is a meaningful paper, which has a certain research value in LiDAR Imaging System. Generally speaking, this article is clear in thinking and obvious in point of view. The paper is well-written but there are some minor problems with the article. Please correct them.
1. Page 1, line 6 (Abstract), " SPAD" is not defined in the text when they are used for the first time.
2. Page 4, line 145, " MCU " is not defined in the text when they are used for the first time.
3. Page 6, line 203, “Ptarget" may been a typo and should be changed to "Preceived".
4. Page 10, line 316-318, “… a lightweight and cost-effective system ….” is not accurate enough, I suggest you add relevant data for comparison.
Minor editing of English language required
Round 2
Reviewer 2 Report
Please refer to the attached pdf.

The Quality of English Language is fine, but the Sec.2 is not readable at all.
